# The Clinical and Epidemiological Profile of Paediatric-Onset Multiple Sclerosis in Poland

**DOI:** 10.3390/jcm11247494

**Published:** 2022-12-17

**Authors:** Waldemar Brola, Barbara Steinborn, Marek Żak, Maria Mazurkiewicz-Bełdzińska, Sergiusz Jóźwiak, Piotr Sobolewski, Maciej Wilski, Małgorzata Bilska, Magdalena Siedlarska, Iwona Puzio-Bochen, Agnieszka Wencel-Warot, Małgorzata Lemka, Sławomir Kroczka, Elżbieta Czyżyk, Małgorzata Bocheńska, Ewa Emich-Widera, Jerzy Pietruszewski, Leszek Boćkowski, Katarzyna Kapica-Topczewska, Agata Czarnowska, Alina Kułakowska, Barbara Ujma-Czapska, Agata Gruna-Ożarowska, Łukasz Przysło, Katarzyna Połatyńska, Magdalena Dudzińska, Krystyna Mitosek-Szewczyk, Aleksandra Melnyk, Monika Adamczyk-Sowa, Katarzyna Kotulska

**Affiliations:** 1Collegium Medicum, Jan Kochanowski University, 25-369 Kielce, Poland; 2Department of Developmental Neurology, Poznan University of Medical Sciences, 61-545 Poznań, Poland; 3Department of Developmental Neurology, Medical University of Gdańsk, 80-952 Gdańsk, Poland; 4Department of Child Neurology, Medical University of Warsaw, 02-091 Warsaw, Poland; 5Department of Adapted Physical Activity, Poznań University of Physical Education, 61-871 Poznań, Poland; 6Department of Neurology and Epileptology, The Children’s Memorial Health Institute, 04-783 Warsaw, Poland; 7Department of Child and Adolescent Neurology, Medical College, Jagiellonian University in Kraków, 30-663 Kraków, Poland; 8Clinical Department of Child Neurology, Clinical Central Hospital No 2 in Rzeszow, 35-301 Rzeszów, Poland; 9Department of Pediatric Neurology, School of Medicine in Katowice, Medical University of Silesia Katowice, 41-808 Katowice, Poland; 10Department of Pediatric Neurology and Rehabilitation, Medical University of Białystok, 15-274 Białystok, Poland; 11Department of Neurology, Medical University of Białystok, 15-276 Bialystok, Poland; 12Department of Social Pediatrics, Faculty of Health Sciences, Wrocław Medical University, 51-618 Wrocław, Poland; 13Department of Developmental Neurology, Polish Mother’s Memorial Hospital Research Institute, 93-338 Łódź, Poland; 14Children’s Neurology Ward, Dr. E. Hanke Centre of Pediatrics and Oncology of Chorzów, 41-500 Chorzów, Poland; 15Department of Child Neurology, Medical University, 20-093 Lublin, Poland; 16Department of Child Neurology, Regional Specialized Children’s Hospital, Olsztyn, Poland, and Collegium Medicum, University of Warmia and Mazury in Olsztyn, 10-082 Olsztyn, Poland; 17Department of Neurology, Faculty of Medical Sciences in Zabrze, Medical University of Silesia in Katowice, 40-055 Katowice, Poland

**Keywords:** paediatric-onset multiple sclerosis, clinical features, childhood-onset, adolescent-onset, prevalence

## Abstract

Background. Paediatric-onset MS (POMS) has a unique clinical profile compared to the more prevalent adult-onset MS. For this study, we aimed to determine the demographic and clinical characteristics of POMS in Poland as well as addressing some of its epidemiological aspects. Methods. A retrospective study was conducted based on the Polish Multiple Sclerosis Registry, considering a population of children and adolescents with MS (age ≤ 18 years). Data were collected by all 13 centres across Poland specializing in diagnosing and treating POMS. The actual course of the disease and its clinical properties were compared between child (≤12 years) and juvenile (>12 years) patients. MS onset and its prevalence were assessed at the end of 2019, stratified by age range. Results. A total of 329 paediatric or juvenile patients (228 girls, 101 boys) with a clinically definite diagnosis of MS, in conformity with the 2017 McDonald Criteria, were enrolled. For 71 children (21.6%), the first symptoms appeared before the age of 12. The female: male ratio increased with age, amounting to 1:1 in the ≤12 years group and to 2.9:1 in the >12 years group. In most cases, the disease had multi-symptomatic onset (31.3%), and its course was mostly of a relapsing–remitting character (95.7%). The initial Expanded Disability Status Score for both groups was 1.63 ± 1.1, whereas the annual relapse rate was 0.84 during the first 2 years. The time between the onset of symptoms and diagnosis was longer in the younger patients (8.2 ± 4.2 vs. 4.6 ± 3.6 months; *p* < 0.005). On 31 December 2019, the age-adjusted prevalence standardized to the European standard population was 5.19/100,000 (95% CI, 4.64–5.78). Significantly higher prevalence was noted in the 13–18 years group (7.12; 95% CI, 6.64–7.86) than in the 9–12 years group (3.41; 95% CI, 2.98–3.86) and the <9 years group (0.56; 95% CI, 0.46–0.64; *p* < 0.001). Conclusion. POMS commencing at the age of ≤12 years is rare, differing significantly from the juvenile-onset and adult MS in terms of clinical characteristics, course, and incidence, as stratified by gender.

## 1. Introduction

Multiple sclerosis is a disease primarily affecting young adults. The first symptoms appear usually between the age of 20 and 40. The disease may also appear at a younger age, however, as well as in individuals over 50 years of age. At present, paediatric MS, defined as multiple sclerosis with onset under the age of 18 (sometimes under the age of 16), is being diagnosed more and more often [1]. The number of patients suffering from multiple sclerosis worldwide exceeds approx. 2.8 million, out of whom 800,000 live in Europe [2]. Even though the epidemiology of POMS is not yet fully understood, pertinent data from the respective medical centres tackling this issue across the world are freely available. It has been estimated that symptoms might appear by the age of 16 in 3–5% of patients, whereas it may appear under the age of 10 for less than 1% of patients [3].

Despite its essential similarities with adult MS, POMS presents many distinguishing characteristics, and the actual course of the disease is different from that affecting adults [4]. It is not entirely clear whether the incidence of MS in children has increased recently, or whether—in view of better diagnostic methods and advances in medical science—MS is more often diagnosed in this particular population. Due to its rare incidence, the overall body of knowledge on POMS has relied mostly on case reports.

Following the establishment of the International Paediatric MS Study Group (IPMSSG) in 2005, the overall scope of research on childhood MS was significantly boosted [5,6]. Better understanding of environmental and genetic risk factors as well as a major rethink of diagnostic criteria significantly enhanced the actual methods for differential diagnosis. The newly available treatment options have also contributed to increased interest in POMS [7].

It has been estimated that the global incidence of POMS ranges from 0.05 to 2.85 per 100,000 children a year, and its prevalence is 0.7 to 26.9 per 100,000 children [8]. As we have managed to demonstrate in our previous study, the prevalence in Poland was 5.19/100,000, and the average annual incidence of POMS from 2015–2019 was estimated at 0.77/100,000 person-years [9].

The aim of our current study is to determine the demographic and clinical characteristics of patients affected by POMS in Poland in due consideration of the differences between patients affected by childhood (≤12 years), juvenile (>12 years), and adult MS as well as estimating its prevalence in specific age groups.

## 2. Materials and Methods

### 2.1. Study Population and Design

As of 31 December 2019, the total number of children and adolescents (aged ≤18 years) was 6,948,706 individuals (3,566,972 boys and 3,381,734 girls) in Poland [10]. Data regarding patients affected by POMS were collected from all 13 Polish medical centres involved in diagnosing and treating MS through the Polish Multiple Sclerosis Registry (RejSM) website (http://www.rejsm.pl).

The study included children and adolescents diagnosed with multiple sclerosis prior to 31 December 2019. In order to assess the frequency of clinical symptoms and the course of the disease during the pre- and post-puberty periods, the patients were divided into two groups: below and above the age of 12. The disease was diagnosed in conformity with the 2017 McDonald criteria [11].

The differential diagnosis of POMS, along with other acquired demyelinating syndromes of the CNS in children (i.e., ADEM, MOG antibody disease, optic neuritis, transverse myelitis, and neuromyelitis optica), warranted extra attention.

The patients who met the criteria for diagnosis of MS were registered in each of the medical centres. The scope of collected data comprised age, gender, place of residence, family history, and pertinent data related directly to the disease, including date and type of initial symptoms, date of diagnosis, disease form, comorbidities, relapses, additional tests (MRI, cerebrospinal fluid test, evoked potentials), and type of treatment (modifying the course of the disease and symptoms and treatment of relapses). The extent of individual disability was assessed in line with the *Extended Disability Status Scale*.

Child neurology specialists experienced in diagnosing POMS were tasked with data collection. In the year preceding the prevalence date, all patients were examined to have their diagnosis verified in conformity with the McDonald criteria, and subsequently, their eligibility for attending the study protocol was confirmed.

### 2.2. Statistical Analysis

Demographic and clinical characteristics of the study population are described using the counts and relative frequencies for the categorical variables and means ± SDs (or medians and interquartile ranges) for numeric variables.

Prevalence was based on the number of paediatric-onset MS patients registered in the RejSM who remained Polish residents on the prevalence date of 31 December 2019. Prevalence is expressed as the number of all cases on the prevalence day divided by Poland’s child population on the same day (available in the Demographic Yearbook of Poland 2020). Crude sex- and age-specific prevalence was calculated as the number of cases on the prevalence day per 100,000 children aged ≤18 years. The prevalence of MS was adjusted by a direct method, making use of the Polish and European populations as standards.

The 95% confidence intervals (CIs) for prevalence were calculated using the Poisson distribution. Comparisons among respective patient cohorts were completed by making use of one-way ANOVA, followed by independent *t*-test for continuous variables and by the Kruskal–Wallis test followed by the Mann–Whitney U-test for ordinal variables. Chi-square test or Fisher’s exact test were used for categorical variables, as appropriate. *p*-values of <0.05 were considered statistically significant. All statistical analyses were carried out with the aid of the STATISTICA software package, version 8.0 (2007; StatSoft, Inc., Tulsa, OK, USA).

### 2.3. Ethical Approval

The participants under 16 years of age were considered eligible for attending the study protocol after pertinent written consent had been furnished by their parents (or next-of-kin). The study protocol was approved and endorsed by the Regional Medical Ethics Review Committee, The Swietokrzyskie Medical Council, Kielce, Poland (Consent Ref. No.10/2011).

## 3. Results

On the prevalence day (31 December 2019), 329 POMS patients (228 girls and 101 boys) were identified with a clinically definite diagnosis of MS in conformity with the McDonald Criteria. The mean disease onset was 14.8 ± 3.2 years. In most cases (78.4%), the disease commenced after the age of 12. The overall ratio of girls to boys was 2.4:1, which was lower in the ≤12-year age group (1:1) than in the >12-year age group (2.9:1). The initial Expanded Disability Status Score for both groups was 1.63 ± 1.1, while the annual relapse rate was 0.842 within the first 2 years. The time between the onset of symptoms and diagnosis was longer in the case of the younger patients (8.2 ± 4.2 vs. 4.6 ± 3.6 months; *p* < 0.005). The disease was most often subject to a relapse–remission pattern (95.7%). The demographic and clinical characteristics of the study patients are provided in Table 1.

The onset of the disease was most often multi-symptomatic (31.3%), associated with visual disturbances (24%) or motor dysfunctions (15.5%). The frequency of specific initial symptoms was similar in the ≤12-year and the 13–18-year age groups.

Overall crude prevalence in Poland’s children population on the prevalence date (31 December 2019) was 4.71/100,000 (95% CI, 4.21–5.24). A significantly higher prevalence was recorded among the girls (6.71; 5.87–7.64) than the boys (2.8; 2.28–3.41; *p* < 0.001). The age-adjusted prevalence standardized to the European standard population was 5.19 per 100,000 (95% CI, 4.64–5.78). Table 2 gives the age- and sex-specific prevalence per 100,000 population on 31December 2019. The age-specific prevalence peaked in the 13–18-year age group. In the cohort under study, there were no disease cases under the age of 5, and in the 5–8-year age group, the prevalence was 0.56/100,000 (95% CI, 0.46–0.64), increasing over the subsequent years and reaching the level of 7.12 (95% CI, 6.64–7.86) in the 13–18-year age group.

In the case of children and adolescents with MS, various comorbidities were also encountered. Most often, these included other autoimmune diseases (e.g., Hashimoto’s disease, autoimmune hepatitis, Goodpasture syndrome, and psoriasis), which were established in over 6% of patients with POMS, especially children <12 years of age. Table 3 presents all additional diseases stratified by respective age groups.

On 31 December 2019, 269 children (82.3%) had been treated with disease-modifying therapy (DMT). A total of 255 children were allocated first-line treatment, and the most commonly applied medication was dimethyl fumarate (43.1%), followed by interferon beta-1a 44 μg (23.9%) and interferon beta-1b (23.1%). Second-line treatment was received by 14 individuals (12 received natalizumab, 1 alemtuzumab, 1 fingolimod). Off-label treatment (rituximab, cyclophosphamide, azathioprine, and intravenous immunoglobulin) was used in 13 rapidly developing severe MS cases (Table 4).

## 4. Discussion

Multiple sclerosis in children and adolescents is a rare and still not yet adequately understood disease. There is a scarcity of epidemiological data regarding paediatric-onset MS; therefore, the clinical characteristics and the actual course of the disease require more extensive research and observation [12].

The Atlas of MS 2020 estimates that MS affects approximately 30,000 children and adolescents worldwide. The authors noted, however, that the available data are most likely under-rated, as they were only sourced from estimates furnished by a dozen or so countries [2]. In Poland, monitoring of the epidemiological situation with regard to MS has only been recently initiated [13]. Studies carried out in recent years have indicated the prevalence in adults as approximately 131/100,000 and the incidence to be 6.6/100,000 person-years [14]. It has been estimated that approximately 50,000 individuals suffer from the disease in Poland.

This study focused on a paediatric cohort, identifying 329 patients whose disease commenced during childhood or adolescence. The prevalence rate standardized to the European standard population on the day of prevalence (i.e., 31 December 2019) was 5.19 per 100,000 (95% CI, 4.64–5.78); notably, a significantly higher prevalence was recorded among girls (7.41; 6.48–8.44) than boys (3.08; 2.5–3.74; *p* < 0.001).

The disease occurred much less often prior to the age of 12. No case was found under the age of 5, and between the age of 5 and 12, the first symptoms appeared in only 71 of the children (21.6%).

It is generally believed that the onset of MS before puberty is extremely rare, as it is particularly hard to diagnose, mostly owing to its atypical presentation [15]. In the group of children aged 5–8, MS was diagnosed in the case of 7 patients, and the prevalence was at 0.56 (95% CI, 0.46–0.64); meanwhile, in the 9–12-year age group, there were 64 cases, and the prevalence was at 3.41 (95% CI, 2.98–3.86).

Results similar to ours have been reported in Germany, where, in the years 2009–2011, there was a distinct (almost 30-fold) increase in incidence from the ≤10-year age group to the 14–15-year age group [16]. All cases comprised a relapse–remission form, with a multi-symptomatic onset in half of the cases—much like in the present study.

Globally, the prevalence ranges from 0.69–26.92 per 100,000 of the paediatric population [8,17]. The highest prevalence rate was recorded in Sardinia, at 26.92 (26.61–27.23), among children and adolescents <18 years of age in 2001–2012 [18]. It would appear that such high prevalence rates might originate from the specific genetic and environmental factors in Sardinia. In other countries, the prevalence is much lower. In an American study of 276 individuals affected by POMS, carried out in 2008–2012, the prevalence rate in children and adolescents <19 years of age was established as 10.41 (9.25–11.71) [19]. In 2013 in Kuwait, Alroughani estimated the prevalence at 6.0 (4.2–8.5) [20]. A study in Abu Dhabi in a population <19 years of age showed a prevalence of 6.54 (4.00–10.6) [21]. The lowest prevalence, of 0.69 (0.58–0.80), was recorded in Japan [22].

In the pre-puberty population, the number of disease cases among girls and boys tends to be relatively equal [15]. Among the adolescents, there was a more significant female domination, and the female:male ratio increased to 2–3:1, which may imply that the onset of menstruation plays a role in the pathogenesis of multiple sclerosis [4].

Our own study demonstrated that MS commencing in adolescence (>12 years of age), similarly to MS among adults, was more common for girls than boys. In the entire <18-year age population, the female:male ratio was 2.4:1; meanwhile, in the >12-year age group, it was 2.9:1, and in the ≤ 12-year age group, it was 1:1.

This was in line with observations carried out in other countries, where similar ratios have been reported [6,21,23,24].

Furthermore, in Denmark—as evidenced by the Danish Multiple Sclerosis Registry—it has been established that MS is extremely rare before puberty, but its incidence significantly increased above the age of 9 in girls and above the age of 11 in boys. The ratio of women to men was at 2.5; and the average incidence age was 16 years. Therefore, it was evidenced that the incidence among children under the age of 18 was four times higher than that among children under 14 years of age [25].

In our own study, the average age of incidence was 14.8 years (range 5–18), and MS at the age of <12 years was more than five times less frequent than between 13 and 18 years of age.

We established a family history of MS among 7.9% of our patients. Other studies have established a family history of MS incidence in 2–14% of the population under study [3,25].

Generally, juvenile-onset MS is similar to adult-onset MS. This said, MS commencing in childhood (<12 years) differs quite significantly from both of these forms [23]. Most POMS follows a relapsing–remitting course (95.7%), while progressive forms are extremely rare. Primary progressive MS was diagnosed in only one girl aged 17.2 years, while in another girl, conversion to secondary progressive form was found after more than eight years of disease (101 months from diagnosis). 

It is especially hard to differentiate between early-childhood MS and other acquired demyelinating syndromes of the CNS in children (i.e., ADEM, optic neuritis, transverse myelitis, and neuromyelitis optica).

Diagnosing a specific POMS often requires much more extensive observation periods as well as extra-prudent differential diagnosis [26]. In our own study, the time between the onset of symptoms and diagnosis was longer in the case of younger patients (8.2 ± 4.2 vs. 4.6 ± 3.6 months; *p* < 0.005).

MS with onset at age ≤ 12 was characterized by more numerous T2 lesions and gadolinium-enhancing lesions in brain MRI than that at an older age (15.4 ± 5.2 vs. 11.2 ± 5.4 and 2.8 ± 1.6 vs. 1.8 ± 1.2, respectively; *p* < 0.005). Despite the large number of changes in MRI and the initially high disability in EDSS, children showed a faster improvement rate than adults, often followed by a complete relief of symptoms. This remains in line with previous observations that, despite severe relapses, symptoms usually remitted, whereas the compensatory capacity of the nervous system in young patients effectively counteracted the rapid progression of disability [27,28]. Although the time required to approach the threshold of damaging motor neurons—consequently resulting in irreversible disability—is approx. 10 years longer for children and adolescents affected by MS than for adults with MS, disability tends to occur at a younger age than for adults with MS [27].

The incidence of oligoclonal bands (OCB) in the cerebrospinal fluid during the pre-puberty period is lower than in older children and adults. In our own study, we encountered OCB in 86% of children ≤12 years of age and in 90% of children >12 years of age. As was noted by McKay et al., in the case of POMS, OCB was more common than among adults although it is not associated with the frequency of relapses and higher EDSS [29].

We also noted the comorbidities, especially other autoimmunological diseases, established in 12.7% of children ≤12 years of age and in 4.3% of children >12 years of age. The differences were significant when compared to the adult-onset MS, where the most common were hypertension (4.3%) and thyroid diseases (3.3%) [30].

Of all the children under study, 82% remained on disease-modifying medications. This is a much higher proportion of patients than in the case of the adult population in Poland (40%). The most commonly administered medication was dimethyl fumarate (43.1%). In terms of second-line treatment, natalizumab was used most often, as other drugs in 2019 were not yet subject to National Health Service reimbursement policy in Poland. Interestingly enough, there was a high percentage of off-label therapies applied in the cases of rapidly developing severe conditions (e.g., rituximab, cyclophosphamide, azathioprine, and intravenous immunoglobulin). The use of the above-mentioned immunosuppressive drugs in children and adolescents was related to the fact that, in the analysed period before 2020, if natalizumab (the only reimbursed drug) failed, there were no other treatment options. In the absence of applicable recommendations, the managers of some of the centres in the case of rapidly developing severe POMS sometimes had to make various controversial therapeutic decisions. 

## 5. Conclusions

POMS commencing before the age of 12 is rare and differs significantly from both juvenile-onset and adult MS in terms of its clinical characteristics, the actual course of the disease, and its incidence as stratified by gender. Before puberty, boys suffer from the disease with a similar frequency as girls; subsequently, its dominance in the female gender is clearly manifest. The onset of the disease at a younger age is associated with a longer time of differential diagnosis as well as a delay in diagnosis and commencement of DMT treatment. Early childhood MS is characterized by more numerous T2 lesions and gadolinium-enhancing lesions in MRI than at an older age. The number of relapses is also greater than among adults, but despite the large number of changes concerning MRI and high EDSS, children tend to show faster improvement rates and often a complete relief of symptoms. In the vast majority of cases among children and adolescents, MS is characterised by a relapsing–remitting course.

## Figures and Tables

**Table 1 jcm-11-07494-t001:** Demographic and clinical characteristics of patients with paediatric-onset MS.

Characteristics	Age ≤ 12 y (*n* = 71)	Age > 12 y (*n* = 258)	Overall (*n* = 329)
Female, *n* (%)	36 (50.7)	192 (74.4)	228 (69.4)
Male, *n* (%)	35 (49.3)	66 (25.6)	101 (30.6)
Female-to-male ratio	1:1	2.9:1	2.4:1
Age (years) at disease onset (mean ± SD; median, (interquartile range))	11.4 ± 2.4; 11	16.7 ± 3.1; 17	14.8 ± 3.2; 15
	(9–12)	(14–18)	(12–17)
Time (months) from first symptoms to diagnosis (mean ± SD (range))	8.2 ± 4.2 (0–48)	4.6 ± 3.6 (0–62)	7.8 ± 13.3 (0–62)
Disease course sub-types, *n* (%)			
Relapsing–remitting	68 (95.8)	247 (95.7)	315 (95.7)
Primary progressive	0	1 (0.4)	1 (0.3)
Secondary progressive	0	1 (0.4)	1 (0.3)
CIS (clinically isolated syndrome)	3 (5.3)	9 (3.3)	12 (3.7)
EDSS score at disease onset (mean ± SD)	1.24 ± 0.8	1.86 ± 1.3	1.63 ± 1.1
Family history of MS, *n* (%)	11 (19.3)	14(5.2)	26 (7.9)
Findings			
Brain MRI, *n* (%)	71 (100)	258 (100)	329 (100)
Number of T2 lesions, mean ± SD	15.4 ± 5.2	11.2 ± 5.4	13.3 ± 4.8
Number of gadolinium-enhancing lesions, mean ± SD	2.8 ± 1.6	1.8 ± 1.2	2.1 ± 1.2
Cerebrospinal fluid, %	95.4	97.2	96.6
Presence of oligoclonal bands	86	90	88
Visual evoked potentials, %	68.2	78.9	75.3
Abnormal	58.6	68.6	62.3

**Table 2 jcm-11-07494-t002:** Age- and sex-specific prevalence (per 100,000) of paediatric-onset multiple sclerosis patients in Poland, current as of 31 December 2019 ^a^.

Age (years)	Male Population				Female Population				Total Children Population			
	Cas.	Population	Crude (95% CI)	ASR (95% CI)	Cas.	Population	Crude (95% CI)	ASR (95% CI)	Cas.	Population	Crude (95% CI)	ASR (95% CI)
**0–4**	0	985,549	-	-	0	932,915	-	-	0	1,918,464	-	-
**5–8**	3	752,477	0.22 (0.14–0.34)	0.28 (0.18–0.38)	4	697,669	0.68 (0.48–0.76)	0.78 (0.64–0.92)	7	1,450,146	0.48 (0.36–0.58)	0.56 (0.46–0.64)
**9–12**	32	734,753	1.84 (1.52–1.96)	1.98 (1.72–2.22)	32	684,724	4.12 (3.74–4.36)	4.42 (3.86–5.12)	64	1,419,477	3.24 (2.84–3.46)	3.41 (2.98–3.86)
**13–18**	66	1,094,193	4.22 (3.84–4.46)	4.36 (4.12–4.52)	192	1,066,426	9.26 (8.74–10.12)	9.76 (9.22–10.18)	258	2,160,619	6.72 (6.12–7.14)	7.12 (6.64–7.86)
**0–18**	**101**	**3,566,972**	**2.80** **(2.28–3.41)**	**3.08** **(2.5–3.74)**	**228**	**3,381,734**	**6.71** **(5.87–7.64)**	**7.41** **(6.48–8.44)**	**329**	**6,948,706**	**4.71** **(4.21–5.24)**	**5.19** **(4.64–5.78)**

Note: ^a^ Based on the data provided by Polish Central Statistical Office; CIs, confidence intervals; ASR, age-standardized rate for European standard population.

**Table 3 jcm-11-07494-t003:** Comorbidities of the patients with POMS.

Comorbidities	Age ≤ 12 y (*n* = 71)	Age > 12 y (*n* = 258)	Overall (*n* = 329)
Other autoimmunological diseases *, *n* (%)	9 (12.7)	11 (4.3)	20 (6.1)
Depression, anxiety, fatigue, *n* (%)	7 (9.8)	8 (3.1)	15 (4.6)
Congenital defects, *n* (%)	7 (9.8)	6 (2.3)	13 (3,9)
Inhalation allergy, *n* (%)	7 (9.8)	4 (1.6)	11 (3.3)
Epilepsy, *n* (%)	5 (7.1)	4 (1.6)	9 (2.7)
Obesity, *n* (%)	3 (4.2)	5 (1.9)	8 (2.4)
Bronchial asthma, *n* (%)	3 (4.2)	3 (1.2)	6 (1.8)
Arterial hypertension, *n* (%)	2 (2.8)	3 (1.2)	5 (1.5)
Diabetes, *n* (%)	2 (2.8)	2 (0.8)	4 (1.2)
Migraine, *n* (%)	1 (1.4)	3 (1.2)	4 (1.2)
Spine defects (scoliosis, discopathy), *n* (%)	2 (2.8)	2 (0.8)	4 (1.2)
Other, *n* (%)	4 (5.6)	8 (3.1)	12 (3.6)
No comorbidities, *n* (%)	24 (33.1)	217 (84.1)	241 (73.3)

Note: * Hashimoto’s disease, autoimmune hepatitis, Goodpasture syndrome, psoriasis.

**Table 4 jcm-11-07494-t004:** List of DMDs in use for POMS between 2015 and 2019.

Disease-Modifying Drugs	First-Line			Second-Line	Off-Label Treatment	Number of Cases
	1	2	3			
Glatiramer acetate	28	19	1	-	Rituximab	3
Dimethyl fumarate	110	41	3	-		
IFN-beta 1a					Cyclophosphamide	1
22 or 44 µg	61	3	-	-		
30 µg	30	2	2	-	Sirolimus	1
Peginterferon beta-1a	5	-	-	-		
IFN-beta 1b 250 µg	59	2	3	-	Azathioprine	1
Teriflunomide	1	-	-	-		
Natalizumab	-	-	-	12	Immunoglobulin	7
Alemtuzumab	-	-	-	1		
Fingolimod	-	-	-	1		

## Data Availability

The data that support the findings of this study are available at RejSM website at http://www.rejsm.pl.

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
