# Peer review of "The Clinical and Epidemiological Profile of Paediatric-Onset Multiple Sclerosis in Poland"

_jcm, 2022, doi:10.3390/jcm11247494_

Round 1

Reviewer 1 Report

The study aimed to determine the epidemiological aspects as well as demographic and clinical characteristics of 329 Polish patients affected by multiple sclerosis during childhood or adolescence, including differences between those affected by the childhood (≤12 years) or  juvenile (>12 years).  The study is well written and presents the topic in a concise manner. I have only some comments.

1.       The name of the country in which the study was conducted should be included in the title of the manuscript.  

2.       The abstract lacks information on which McDonald's criteria were used to establish MS diagnosis.

3.       Does the number of of T2 lesions as well as gadolinium-enhancing lesions in Table 1 apply only to the brain or also to the spinal cord?

4.       There is no information on disease duration. This is especially important when analyzing conversion to SPMS. As only one person experienced the transition to SPMS, I'm curious how long after the onset of the disease this occurred?

5.       It would be interesting to add information whether progressive POMS variants  have been found in boys or girls?

6.   Off-label treatment in rapidly developing severe POMS, especially with sirolimus, azathioprine or cyclophosphamide in children and adolescents, requires further explanation. Was it also due to comorbidities? I am really surprised that azathioprine is given as a drug for rapidly developing severe  POMS. Please comment more.

Author Response

Response to Reviewer 1 Comments

 We feel much indebted to the Reviewer for all his insightful suggestions and recommendations, as this appreciably helped us out in revising the manuscript. Our responses, addressing on a point-by-point basis all the issues raised by the Reviewer, are to be found further below.

Comments and Suggestions for Authors

The study aimed to determine the epidemiological aspects as well as demographic and clinical characteristics of 329 Polish patients affected by multiple sclerosis during childhood or adolescence, including differences between those affected by the childhood (≤12 years) or  juvenile (>12 years).  The study is well written and presents the topic in a concise manner. I have only some comments.

Point 1.  The name of the country in which the study was conducted should be included in the title of the manuscript.  

Response 1: Thank you very much for your valuable comment. We changed the title of the manuscript as suggested: „The clinical and epidemiological profile of paediatric-onset multiple sclerosis in Poland”

Point 2. The abstract lacks information on which McDonald's criteria were used to establish MS diagnosis.

Response 2: We added in abstract following information: A total of 329 paediatric or juvenile patients (228 girls, 101 boys) with a clinically definite diagnosis of MS, in conformity with the 2017 McDonald Criteria, were enrolled.

Point 3. Does the number of of T2 lesions as well as gadolinium-enhancing lesions in Table 1 apply only to the brain or also to the spinal cord?

Response 3: The number of T2 lesions and gadolinium-enhancing lesions apply only to the brain. We added in Table 1 record: „Brain MRI” 

Point 4. There is no information on disease duration. This is especially important when analyzing conversion to SPMS. As only one person experienced the transition to SPMS, I'm curious how long after the onset of the disease this occurred?

Point 5. It would be interesting to add information whether progressive POMS variants  have been found in boys or girls?

Response 4 and 5: Thank you for your valuable remarks. In reference to both of the above-mentioned points, in the "Discussion" section, we discussed this topic a bit more extensively, adding an explanation:

Most POMS follows a relapsing–remitting course (95.7%), while progressive forms are extremely rare. Primary progressive MS was diagnosed in only one girl aged 17.2 years; while, in another girl, conversion to secondary progressive form was found after more than eight years of disease (101 months from diagnosis).

Point 6. Off-label treatment in rapidly developing severe POMS, especially with sirolimus, azathioprine or cyclophosphamide in children and adolescents, requires further explanation. Was it also due to comorbidities? I am really surprised that azathioprine is given as a drug for rapidly developing severe  POMS. Please comment more.

Response 6:

The authors of the article are also surprised by some off-label drugs (sirolimus, azathioprine, or cyclophosphamide). Although these were only isolated cases, it is difficult to rationally justify the use of the above-mentioned drugs, especially without mentioning the accompanying diseases. However, the POMS treatment centers reported such treatment, and the authors had to include it in the analysis. During the study, the only reimbursed drug in severe POMS was natalizumab, and in the event of its ineffectiveness, there were no other treatment options. In the absence of applicable recommendations, clinical centers sometimes made controversial therapeutic decisions.

At the end of the "Discussion" section, a paragraph was added:

The use of the above-mentioned immunosuppressive drugs in children and adolescents was related to the fact that, in the analysed period before 2020, if natalizumab (the only reimbursed drug) failed, there were no other treatment options. In the absence of applicable recommendations, the managers of some of the centres in the case of rapidly developing severe POMS sometimes had to make various controversial therapeutic decisions.

Reviewer 2 Report

The presented epidemilogical article is about the highly important topic of pediatric onset MS. It is a well conceived study, carried out with adequate statistical methods. The results are well presented and discussed and are of great interest to the readers. 

Author Response

Response to Reviewer 2 Comments

The presented epidemilogical article is about the highly important topic of pediatric onset MS. It is a well conceived study, carried out with adequate statistical methods. The results are well presented and discussed and are of great interest to the readers. 

Response: We wish to thank the Reviewer for a positive review of our manuscript.

Reviewer 3 Report

This manuscript provides important information on pediatric onset MS. The study design is very good in that all sites diagnosing children with MS in Poland are included. The results are in accordance with earlier studies but contributes also new pieces to the puzzle. 

However it is obvious that english is not the author's main language, so extensive editing of English language is required

Author Response

Response to Reviewer 3 Comments

Point 1: This manuscript provides important information on pediatric onset MS. The study design is very good in that all sites diagnosing children with MS in Poland are included. The results are in accordance with earlier studies but contributes also new pieces to the puzzle. 

Response 1: Thank you very much for your valuable comment and positive review of our manuscript.

Point 2: However it is obvious that english is not the author's main language, so extensive editing of English language is required.

Response 2: The manuscript has been linguistically corrected by MDPI Language Editing Services.